# Maximal expiratory pressure is associated with reinstitution of mechanical ventilation after successful unassisted breathing trials in tracheostomized patients with prolonged mechanical ventilation

**Shwu-Jen Lin**[1], **Jih-Shuin Jerng**[2]*, **Yao-Wen Kuo**[1], **Chao-Ling Wu**[1], **Shih-Chi Ku**[2], **Huey-Dong Wu**[1]

**1** Department of Integrated Diagnostics & Therapeutics, National Taiwan University Hospital, Taipei, Taiwan,
**2** Department of Internal Medicine, National Taiwan University Hospital, Taipei, Taiwan

* jsjerng@ntu.edu.tw

## Abstract

### Objective

Reinstitution of mechanical ventilation (MV) for tracheostomized patients after successful weaning may occur as the care setting changes from critical care to general care. We aimed to investigate the occurrence, consequence and associated factors of MV reinstitution.

### Methods

We analyzed the clinical data and physiological measurements of tracheostomized patients with prolonged MV discharged from the weaning unit to general wards after successful weaning to compare between those with and without in-hospital MV reinstitution within 60 days.

### Results

Of 454 patients successfully weaned, 116 (25.6%) reinstituted MV at general wards within 60 days; at hospital discharge, 42 (36.2%) of them were eventually liberated from MV, 51 (44.0%) remained MV dependent, and 33 (28.4%) died. Of the 338 patients without reinstitution within 60 days, only 3 (0.9%) were later reinstituted with MV before discharge (on day 67, 89 and 136 at general wards, respectively), and 322 (95.2%) were successfully weaned again at discharge, while 13 (3.8%) died. Patients with MV reinstitution had a significantly lower level of maximal expiratory pressure ($P_E$max) before unassisted breathing trial compared to those without reinstitution. Multivariable Cox regression analysis showed fever at RCC discharge (hazard ratio [HR] 14.00, 95% confidence interval [CI] 3.2–61.9) chronic obstructive pulmonary disease (HR 2.37, 95% CI 1.34–4.18), renal replacement therapy at the ICU (HR 2.29, 95% CI 1.50–3.49) and extubation failure before tracheostomy (HR 1.76, 95% CI 1.18–2.63) were associated with increased risks of reinstitution, while $P_E$max > 30 $cmH_2O$ (HR 0.51, 95% CI 0.35–0.76) was associated with a decreased risk of reinstitution.

**Data Availability Statement:** All relevant data are within the manuscript and its Supporting Information files.

**Funding:** The authors received no specific funding for this work.

**Competing interests:** The authors have declared that no competing interests exist.

## Conclusions

The reinstitution of MV at the general ward is significant, with poor outcomes. The $P_E$max measured before unassisted breathing trial was significantly associated with the risk of reinstituting MV at the general wards.

## Introduction

Liberation from mechanical ventilation (MV) in a critical care setting remains challenging. About 10% of patients with acute respiratory failure may require prolonged MV, commonly defined as longer than 21 days [1]. These patients have a very poor prognosis, with a one-year survival rate between 40% and 50% [2]. Prolonged MV also imposes a significant care burden on intensive care units (ICUs) [3]. Successful weaning with sustainable independence from invasive MV is, therefore, pivotal to the management of patients with prolonged MV.

One of the problems encountered during MV liberation is the reinstitution of MV after the patient is transferred out the critical care setting after being deemed successfully weaned by a protocoled process [4,5], commonly based on an operational definition of 5 days of unassisted breathing. A study reported that reinstitutions occurred within 28 days in 52% of patients, indicating that enduring freedom could not be established until > 28 days had elapsed [5]. However, the associated clinical and physiological factors for the reinstitution are uncertain, and few reports have investigated the reinstitution of MV support in successfully weaned patients with prolonged MV [1, 5–7].

Traditionally, observing a patient's clinical respiratory condition is used to define being successfully liberated from the ventilator during the last few days of a continuous unassisted breathing trial (UBT), but without the routine application of regular assessments of respiratory physiologic parameters. Measuring weaning parameters has been commonly applied in the extubation of intubated patients [8]; however, few studies have investigated its application in tracheostomized patients [9], and no studies have reported its use in assessing tracheostomized patients undergoing days of an unassisted breathing trial. While physiologic measurements remain difficult in non-intubated patients, patients with a tracheostomy tube may provide an opportunity to explore the respiratory mechanics during an unassisted breathing trial when clinicians need to assess the feasibility of transferring out the patients who are considered to have been successfully weaned from the ventilator.

In this study, we aimed to investigate whether weaning parameters were associated with the reinstitution of MV in patients who were successfully weaned from a ventilator in a protocoled weaning care setting.

## Materials and methods

### Design and setting

This retrospective study was conducted at the Respiratory Care Center (RCC), a dedicated weaning unit of National Taiwan University Hospital, between January 2016 and December 2018. This Respiratory Care Center has 15 beds and receives patients with prolonged MV from the ICUs of the same hospital. The Research Ethics Committee B of this hospital approved this study (#201902056RINB) and waived the need for informed consent from the patients.

The decision of weaning after admission to the Respiratory Care Center was made by the attending physician. The Respiratory Care Center uses a standardized weaning protocol that is

comparable to that reported by Jubran et al. [4]. Briefly, the ventilator settings were gradually reduced to pressure support of 10 cmH$_2$O and an end-expiratory positive pressure of 5 cmH$_2$O for at least 8 hours to assure the patient's tolerance. After the respiratory therapist had measured the physiological variables, the patient then underwent a screening procedure that consisted of unassisted breathing for 12 hours for 2 consecutive days with humidified oxygen delivered through a T-piece oxygen tube, The patients who did not develop distress during the screening period then underwent 5 consecutive days of unassisted breathing as a direct liberation trial. The patients who failed this screening process were subjected to a stepwise liberation trial, which consisted of daily increases in the duration of unassisted breathing starting from 2 hours, then extending to 2 hours twice daily, 4 hours daily, 4 hours twice daily, 8 hours daily, 12 hours daily, 16 hours daily, 20 hours daily, and finally continuous unassisted breathing for 5 days. If the patient repeatedly failed to complete the session, they underwent a slow weaning trial which consisted of either breathing through a T-piece but supported with an external positive airway pressure device with a gradual reduction in the pressure level, or stepwise liberation as with the stepwise trial, but repeating each duration of unassisted breathing for 3 to 5 days so the care team could assess the feasibility of further weaning with a longer duration of unassisted breathing and more extended time. The patients who tolerated the liberation process and the final 5 days of continuous unassisted breathing trial were transferred out of the Respiratory Care Center to the general ward for further care and preparation for hospital discharge. Those who failed the liberation process in at least two cycles of the liberation trial were transferred to the long-term respiratory care ward or to the general to manage the medical problems in addition to MV.

## Patients

We included patients aged 20 years or older who were admitted to the Respiratory Care Center and were transferred out to a general ward between January 2016 and December 2018 after successful weaning from a ventilator, defined as at least 5 consecutive days of unassisted breathing without reinstitution of MV at the Respiratory Care Center. Patients with at least one of the following conditions were excluded: no data of weaning parameters after tracheostomy measured at the Respiratory Care Center or within 7 days before transfer to the Respiratory Care Center from an ICU; those with reinstitution of MV due to surgery or other elective interventions; those who received high-flow oxygen support or any device providing noninvasive ventilation at a general ward. We did not exclude those who received MV in the operating room during surgical interventions.

## Data collection and variables

The following data were retrieved from the electronic medical records of the hospital: age, gender, dates of hospitalization, ICU and Respiratory Care Center admissions, dates of Respiratory Care Center discharge to a general ward and hospital discharge, in-hospital outcomes including survival status and ventilator liberation status, dates of initiation of MV, tracheostomy, initiation of the first and last session of the unassisted breathing trial weaning process, and reinstitution of MV, co-morbidities, main condition related to MV at the ICU, main intervention related to respiratory failure, acute physiology and chronic health evaluation (APACHE)- II score, number of extubation before tracheostomy, method of weaning for the session with weaning success, and data of weaning parameters. The weaning parameters included maximal inspiration pressure (P$_I$max), maximal expiration pressure (P$_E$max), tidal volume, minute ventilation, respiratory rate, and rapid shallow breathing index (RSBI), and were measured by the respiratory therapist in the care team at the Respiratory Care Center.

Measuring weaning parameters after tracheostomy is part of the usual care process at the Respiratory Care Center, and is routinely performed after the patients have been transferred to the Respiratory Care Center if not done within 7 days before the transfer. If the tracheostomy was performed at the Respiratory Care Center after the patient had been admitted, the measurements were performed within 3 days after the procedure before initiation of the active weaning process. The weaning parameters were measured every 14 days during the Respiratory Care Center stay and also before the patients were transferred to a general ward upon successful weaning from MV depending on the clinical condition of the patients. In this study, as the timing of initiating unassisted breathing trial was determined by clinicians who may not necessarily have recorded the measurements of weaning parameters, we only selected the data of weaning parameters closest to the beginning of unassisted breathing trial. This real-world data may have included measurements of weaning parameters for tracheostomized patients before they were transferred out of the Respiratory Care Center after days of un-interrupted unassisted breathing trial, however the decision to take the measurements was based on the judgement of clinicians and respiratory therapist, who may have provided expert opinions about further care at the general wards. The measurements were performed based on the standards of care by a respiratory therapist, as reported previously [10], including using a device with a unidirectional valve [11]. As only 12 patients had missing weaning parameter data, we decided to exclude them from sensitivity analysis. The primary outcome of this study was in-hospital MV reinstitution within 60 days. A patient was allocated to the 'non-reinstitution' group if the they died in the hospital without reinstitution of MV.

## Statistical analysis

Demographic and clinical characteristics are expressed as mean ± standard deviation for continuous variables and number (percentage) for categorical variables. Comparisons between the reinstitution and non-reinstitution groups were performed using Fisher's exact test for categorical variables, and the independent sample t-test for continuous variables. Comparisons of paired data of weaning parameters were performed using the paired sample t-test. To investigate the associated factors of reinstitution of MV, we performed a multivariable Cox proportional hazard model within 60 days during the hospitalization. Variables entered in the multivariable Cox model were those variables whose significance less than 0.1 in the univariate analyses. To facilitate the clinical use of this multivariable analysis, several continuous variables were dichotomized in the multivariable analysis, including APACHE II score on ICU admission (by 25 scores), APACHE II score on Respiratory Care Center admission and at Respiratory Care Center discharge (by 15 scores) and $P_E$max (by 30 cmH2O). A two-sided $p$-value <0.05 was considered to be statistically significant, and no adjustment of multiple testing (multiplicity) was made in this study. All statistical analyses were performed using SPSS version 22.0 (SPSS, Chicago, IL, USA).

## Results

### Demographic and clinical characteristics and patient outcomes

From January 2016 to December 2018, a total of 1,069 patients were discharged from the Respiratory Care Center. Of them, 620 (58.0%) were successfully liberated from the ventilator and transferred to general wards, including 154 who were successfully extubated from an endotracheal tube and 466 who were liberated with a tracheostomy tube. Of the tracheostomized patients with successful weaning, 454 had at least one set of valid data of weaning parameters, and they constituted the analysis cohort of this study. Fig 1 depicts the patient inclusion flow diagram for this study (Fig 1). Of the included patients, MV was reinstituted

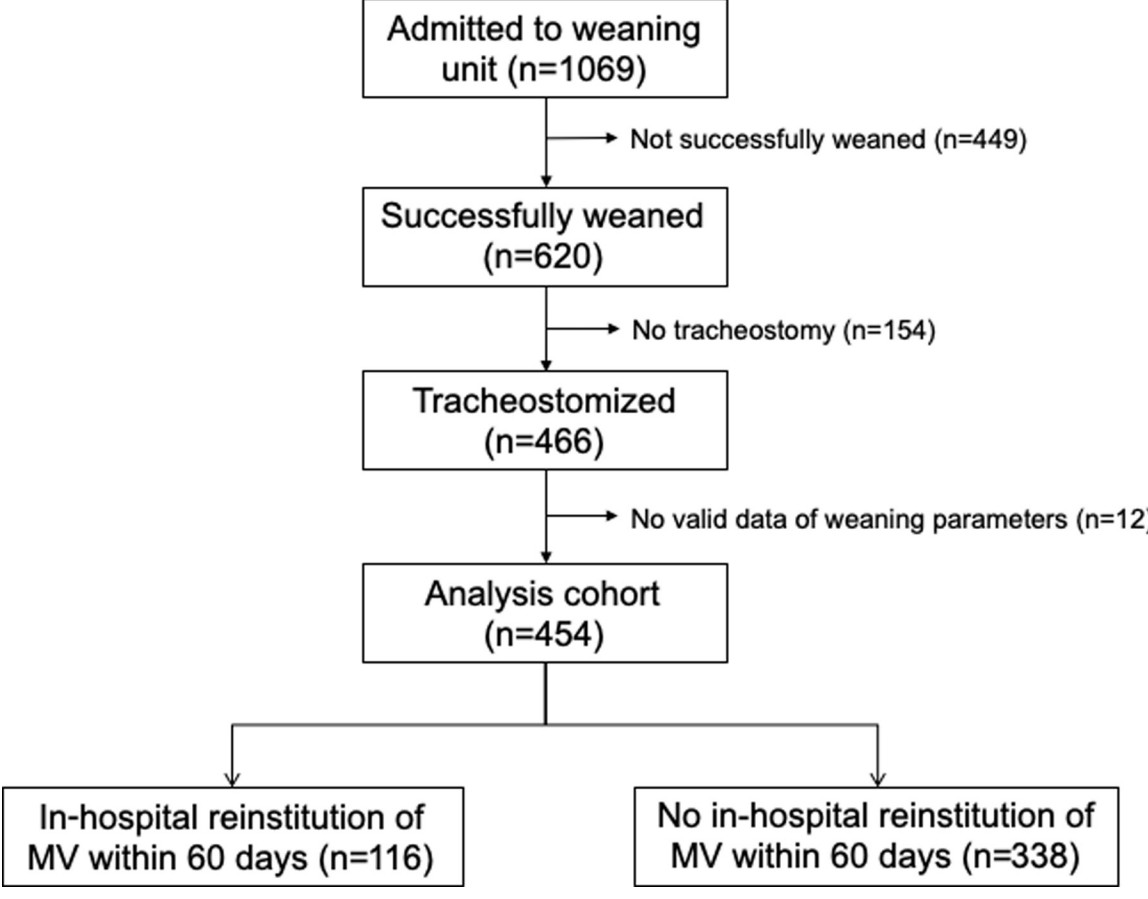

**Fig 1. Flow diagram of patient inclusion and outcomes.**

within 60 days in 116 (25.6%) during their stay at a general ward, with a median interval of 8 days (range, 0–58 days). Of these 116 cases, 57 (49.2%) were reinstituted in less than 7 days after transfer to the general ward, 83 (71.6%) in less than 14 days. T hospital discharge, 42 (36.2%) of these 116 patients were eventually liberated from MV, 51 (44.0%) remained MV dependent, and 33 (28.4%) died. Of the 338 patients without reinstitution of MV within 60 days, only 3 (0.9%) were later reinstituted before discharge (on day 67, 89 and 136 after transfer to general wards, respectively), and 322 (95.2%) were discharged without the need for MV, while 13 (3.8%) died.

Table 1 summarizes the demographic and clinical characteristics of the 454 patients and comparisons between those with and without MV reinstitution. Overall, the patients were generally older, with 64.1% being older than 65 years and 13.9% being older than 85 years. Most of the patients were male, and co-morbidities were common. The most common conditions related to the use of MV in the ICU were pneumonia (55.3%), post-operative status (32.6%), and cerebral hemorrhage/head injury (31.7%). Almost 30% had experienced at least one episode of extubation failure before tracheostomy was performed. Nearly half of the patients were successfully weaned from MV using the direct liberation methods.

Many of the demographic and clinical characteristics were similar between the patients with and without MV reinstitution as shown in Table 1, however significant differences were observed in univariate analysis, including malignancy (p = 0.02), chronic obstructive pulmonary disease (p = 0.01), cerebral hemorrhage/head injury (p = 0.015), renal replacement

**Table 1. Demographics and clinical characteristics of the 454 patients in this study.**

| Characteristic | Total (n = 454) | Reinstitution within 60 days | | p-value |
| --- | --- | --- | --- | --- |
| | | Yes (n = 116) | No (n = 338) | |
| Age, mean±SD | 68.9 ± 16.2 | 71.2 ± 14.8 | 68.1 ± 16.6 | 0.08 |
| Sex, male (%) | 299 (65.9) | 77 (66.4) | 222 (65.7) | 0.89 |
| Co-morbidity | | | | |
| Hypertension | 157 (34.6) | 34 (29.3) | 123 (36.4) | 0.17 |
| Congestive heart failure | 147 (32.4) | 41 (35.3) | 106 (31.4) | 0.43 |
| Diabetes mellitus | 123 (27.1) | 33 (28.4) | 90 (26.6) | 0.70 |
| Neurologic disease | 114 (25.1) | 34 (29.3) | 80 (23.7) | 0.23 |
| Chronic kidney disease | 113 (24.9) | 39 (33.6) | 74 (21.9) | 0.01 |
| Malignancy | 112 (24.7) | 38 (32.8) | 74 (21.9) | 0.02 |
| Liver cirrhosis | 46 (10.1) | 14 (12.1) | 32 (9.5) | 0.42 |
| COPD | 36 (7.9) | 16 (13.8) | 20 (5.9) | 0.01 |
| Main conditions related to MV use | | | | |
| Pneumonia | 251 (55.3) | 73 (62.9) | 178 (52.7) | 0.06 |
| Post-operation MV use | 148 (32.6) | 34 (29.3) | 114 (33.7) | 0.38 |
| Cerebral hemorrhage/head injury | 144 (31.7) | 26 (22.4) | 118 (34.9) | 0.01 |
| Sepsis | 72 (15.9) | 25 (21.6) | 47 (13.9) | 0.05 |
| Heart failure | 48 (10.6) | 15 (12.9) | 33 (9.8) | 0.34 |
| ARDS | 20 (4.4) | 7 (6.0) | 13 (3.8) | 0.32 |
| Spinal cord injury | 12 (2.6) | 5 (4.3) | 7 (2.1) | 0.31 |
| Admitted from medical ICU | 242 (53.3) | 62 (53.4) | 180 (53.3) | 0.97 |
| Intervention at the ICU | | | | |
| Tracheostomy | 444 (97.8) | 115 (99.1) | 329 (97.3) | 0.25 |
| Renal replacement therapy | 88 (19.4) | 38 (32.8) | 50 (14.8) | <0.01 |
| Inhaled nitric oxide | 10 (2.2) | 4 (3.4) | 6 (1.8) | 0.46 |
| Prone positioning | 7 (1.5) | 2 (1.7) | 5 (1.5) | 1.00 |
| ECMO | 6 (1.3) | 3 (2.6) | 3 (0.9) | 0.35 |
| Extubation failure before tracheostomy | 132 (29.1) | 42 (36.2) | 90 (26.6) | 0.05 |
| Extubation times before tracheostomy | 1.32 ± 0.55 | 1.41 ± 0.60 | 1.29 ± 0.53 | 0.05 |
| ICU days before RCC transfer | 25.7 ± 17.0 | 25.6 ± 19.2 | 25.8 ± 16.3 | 0.91 |
| RCC length of stay | 15.5 ± 8.1 | 16.3 ± 8.6 | 15.2 ± 7.9 | 0.21 |
| Total MV days | 38.7±27.9 | 38.8 ± 24.2 | 38.6 ± 29.1 | 0.95 |
| MV days at ICU | 28.6±26.8 | 27.9 ± 23.2 | 28.9 ± 28.0 | 0.73 |
| MV days at RCC | 10.1±8.1 | 10.9 ± 8.7 | 9.8 ± 7.9 | 0.19 |
| APACHE II score on admission to the ICU | 23.9±7.4 | 25.6 ± 7.1 | 23.2 ± 7.4 | <0.01 |
| APACHE II score on admission to the RCC | 14.9 ± 5.0 | 15.7 ± 4.9 | 14.6 ± 5.0 | 0.03 |
| APACHE II score at RCC discharge | 17.0 ± 4.4 | 18.0 ± 4.8 | 16.6 ± 4.2 | <0.01 |
| Heart rate at RCC discharge | 87.0 ± 14.5 | 87.2 ± 15.7 | 86.9 ± 14.1 | 0.88 |
| Respiratory rate at RCC discharge | 20.7 ± 4.7 | 20.6 ± 5.0 | 20.7 ± 4.6 | 0.85 |
| Mean blood pressure at RCC discharge | 88.3 ± 12.4 | 87.8 ± 12.5 | 88.4 ± 12.4 | 0.63 |
| Fever (BT>38.3 degree) at RCC discharge | 2 (0.4) | 2 (1.7) | 0 (0.0) | 0.07 |
| Mode of weaning success | | | | 0.21 |
| Direct liberation | 199 (43.8) | 43 (37.1) | 156 (46.2) | |
| Stepwise protocol | 245 (54.0) | 71 (61.2) | 174 (51.5) | |
| Slow weaning | 10 (2.2) | 2 (1.7) | 8 (2.4) | |
| In-hospital mortality | 47 (10.4) | 33 (28.4) | 14 (4.1) | <0.01 |

APACHE II: acute physiology and chronic health evaluation II; ARDS: acute respiratory distress syndrome; BT: body temperature (Celsius); COPD: chronic obstructive pulmonary disease; ECMO: extracorporeal membrane oxygenation; ICU: intensive care unit; MV: mechanical ventilation; RCC: respiratory care center.

therapy ($p<0.01$), and APACHE II scores on admission to the ICU ($p<0.01$), on admission to the Respiratory Care Center ($p = 0.03$) and at discharge from Respiratory Care Center ($p = 0.004$). The reinstituted group also had a significantly higher in-hospital mortality rate (28.4% vs. 4.1%, $p<0.01$).

## Comparisons of clinical and physiologic variables

Table 2 summarizes the weaning parameter data of the included patients and comparisons between groups. Measurement of parameters was performed 9.7±7.9 days before the first day of continuous unassisted breathing trial. The patients without in-hospital reinstitution of MV within 60 days had similar intervals of parameter measurements and initiation of unassisted breathing trial (9.6±8.0 days vs. 9.9±7.7 days, p = 0.70, data not shown). In general, the 454 patients had satisfactory results before weaning, with 404 (89.0%) having a $P_I$max better than -20 cmH$_2$O, 306 (67.4%) with a $P_E$max better than 30 cmH$_2$O, and 328 (72.2%) with a rapid shallow breathing index better than 105 (refer to Additional S2 File). The patients with MV reinstitution had a significantly higher level of $P_E$max than those without MV reinstitution (43 ±20 vs. 37±17; $p<0.01$). However, there were no significant differences in $P_I$max, tidal volume, minute ventilation, respiratory rate, and RSBI between the two groups (Table 2).

Of the 454 patients, 290 (63.9%) also had additional measurements of weaning parameters upon transfer out to a general ward (S1 Table in S1 File). S2 Table S1 File shows comparisons of weaning parameter data before the unassisted breathing trial and after successful weaning of the 290 patients using the paired t-test. There were no significant evolutional changes in the data for each parameter (S2 Table in S1 File); therefore, we decided to include only weaning parameters before the unassisted breathing trial for consideration in further multivariable regression analysis.

## Multivariable analysis for factors associated with time to reinstitution of MV

Table 3 summarizes the results of multivariable Cox regression analysis, in which the time to reinstitution of MV within 60 days during the hospitalization was the outcome variable. The Cox regression analysis showed that fever at RCC discharge (hazard ratio [HR] 14.00, 95% confidence interval [CI] 3.2–61.9), chronic obstructive pulmonary disease (HR 2.37, 95% CI 1.34–4.18), renal replacement therapy at the ICU (HR 2.29, 95% CI 1.50–3.49) and extubation failure before tracheostomy (HR 1.76, 95% CI 1.18–2.63) were associated with increased risks

**Table 2. Comparisons of weaning parameters between the patients with and without mechanical ventilation reinstitution at a general ward.**

| Parameter | Total (n = 454) | Reinstitution within 60 days | | p-value |
| --- | --- | --- | --- | --- |
| | | Yes (n = 116) | No (n = 338) | |
| $P_I$max, cmH$_2$O | 39 ± 13 | 37 ± 12 | 39 ± 14 | 0.19 |
| $P_E$max, cmH$_2$O | 42 ± 19 | 37 ± 17 | 43 ± 20 | <0.01 |
| $V_T$, mL | 341 ± 124 | 334 ± 110 | 344 ± 128 | 0.49 |
| $V_E$, L | 8.5 ± 3.0 | 8.3 ± 2.6 | 8.5 ± 3.1 | 0.46 |
| RR, breaths/min | 25.8 ± 7.3 | 25.6 ± 6.7 | 25.8 ± 7.5 | 0.78 |
| RSBI, breaths/min/L | 89.3 ± 52.2 | 88.6 ± 48.9 | 89.5 ± 53.3 | 0.88 |

PImax: maximal inspiratory pressure; PEmax: maximal expiratory pressure; VT: tidal volume; VE: minute ventilation volume; RR: respiratory rate; RSBI: rapid shallow breathing index.

**Table 3. Multivariable Cox regression analysis of factors associated with time to reinstitution of MV within 60 days during the hospitalization.**

| Variable | HR (95% CI) | p-value |
|---|---|---|
| Age, year | 1.00 (0.98–1.01) | 0.96 |
| Malignancy | 1.44 (0.95–2.18) | 0.09 |
| COPD | 2.37 (1.34–4.18) | <0.01 |
| Pneumonia | 1.19 (0.74–1.91) | 0.47 |
| Cerebral hemorrhage/head injury | 1.07 (0.61–1.87) | 0.81 |
| Sepsis | 1.09 (0.68–1.75) | 0.72 |
| Renal replacement therapy at ICU | 2.29 (1.50–3.49) | <0.01 |
| Extubation failure before tracheostomy | 1.76 (1.18–2.63) | <0.01 |
| APACHE II score on ICU admission $\geq 25$ | 1.38 (0.93–2.07) | 0.11 |
| APACHE II score on RCC admission $\geq 15$ | 1.40 (0.93–2.10) | 0.10 |
| APACHE II score at RCC discharge $\geq 15$ | 1.30 (0.76–2.22) | 0.35 |
| Fever (BT>38.3 degree) at RCC discharge | 14.00 (3.2–61.9) | <0.01 |
| $P_E$max > 30 cmH$_2$O | 0.51 (0.35–0.76) | <0.01 |

APACHE II: acute physiology and chronic health evaluation II; BT: body temperature; CI: confidence interval; COPD: chronic obstructive pulmonary disease; HR: hazard ratio; MV: mechanical ventilation; ICU: intensive care unit; $P_E$max: maximal expiratory pressure; RCC: respiratory care center.

of reinstitution, while $P_E$max > 30 cmH$_2$O (HR 0.51, 95% CI 0.35–0.76) was associated with a decreased risk of reinstitution.

Fig 2 shows the Kaplan-Meier survival curves of the outcome of in-hospital MV reinstitution within 60 days. Patients with $P_E$max > 30 cmH$_2$O had a significantly lower probability of reinstitution during their stay at a general ward before discharge from the hospital (log-rank test, p<0.01) (Fig 2).

## Discussion

In this study, we found that the tracheostomized patients with prolonged MV had a substantial probability of MV reinstitution at a general ward despite being liberated from a ventilator according to predetermined criteria at a dedicated weaning unit and that these patients had poor outcomes at a general ward. While several clinical characteristics were associated with a higher risk of MV reinstitution at a general ward, we also found that $P_E$max was significantly associated with in-hospital MV reinstitution.

The reinstitution of MV can be a significant issue as it requires modification of the care setting. In this study, the definition of not requiring MV support for at least 5 days to determine successful weaning is compatible with the literature [5]. While patients requiring prolonged MV generally had a poorer long-term prognosis [12], our findings of 26% of reinstitution rate with a higher in-hospital mortality rate (29% vs. 4%) were compatible with previously reported that reinstitution within 14 days was a poor predictor for prolonged MV patients after successful weaning, and physicians should closely monitor such patients [6].

Our finding that $P_E$max consistently served as a significant factor associated with developing a respiratory condition requiring MV reinstitution has not previously been reported in the literature. Although multiple comorbidities and in-hospital clinical condition contribute to an increased risk of reinstituting MV, for tracheostomized patients, there might be a common respiratory condition which increases the risk of further developing respiratory failure. While a previous study suggested that a more normal rapid shallow breathing index and static

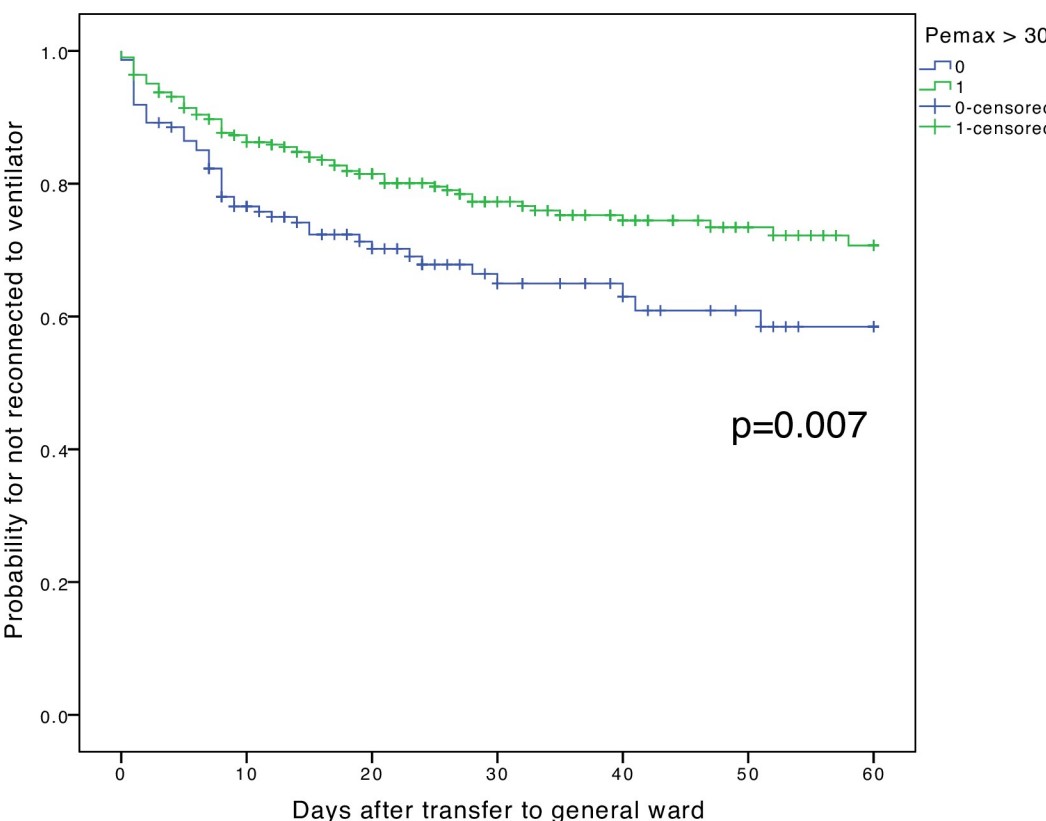

**Fig 2. Kaplan-Meier plot for the reinstitution of MV within 30 days, grouped by before-weaning $P_E$max $>$ 30 cmH$_2$O.**

compliance were associated with better weaning outcome and prolonged survival [13], other researchers showed controversial results [14]. Further reported potential clinical and physiologic predictors for successful weaning from prolonged MV might include hypercapnic ventilatory response [15], duration of stay at the weaning unit, blood urea nitrogen levels, modified Glasgow coma scores, serum albumin, and $P_I$max levels [16], trans-diaphragmatic pressure and tension-time index of the diaphragm [17]. A possible explanation why $P_E$max has rarely been described as a prognostic factor for MV reinstitution may be that most studies on ventilator weaning have focused on extubation success, with weaning parameters being measured before the endotracheally intubated patients proceed with a spontaneous breathing trial for extubation.

Assessments of weaning parameters traditionally focus on $P_I$max and rapid shallow breathing index, which also focus mainly on inspiratory muscle strength and lung mechanics rather than expiratory muscle strength. Our findings suggest that the patients who succeeded in the 5-day unassisted breathing trial assessment generally had adequate inspiratory function, and that the critical factor for the reinstitution of MV at a general ward may have been expiratory strength or cough function, which may not have been identified in clinical observations of breathing condition during the 5-day unassisted breathing trial at the weaning unit. The measurement of $P_E$max in this study conformed to standard procedures as previously validated [10], although the reference value for $P_E$max has not been as frequently discussed in the literature as $P_I$max [18]. As it may have been difficult to perform the measurements due to poor cooperation and respiratory distress during the testing, it is also possible that this suboptimal testing condition suggests a deterioration in their respiratory function such as an inability to

remove lower airway secretions or to cough out aspirated material from the upper airway. PEmax, therefore, may determine the ability of airway clearance, which contributes to the outcome in patients staying in general wards, which are equipped with fewer human resources and monitoring facilities. We also found that the probability of MV reinstitution was not related to the weaning methods in this study. This also suggests that under observation during the 5-day assessment period, the strength of inspiratory muscles may have determined the weaning success in most of the patients without concerns of early transfer out. Our findings may provide further insights into the choice of weaning method based on the literature that unassisted breathing, compared with pressure support, can result in a shorter median weaning time, and that weaning mode does not affect survival at 6 and 12 months [4].

Our finding that $P_E$max contributed to MV reinstitution has several clinical implications. Understanding the presence of suboptimal $P_E$max and lowered reserve during continuous unassisted breathing trial may allow the general ward for the initiation of proactive care processes such as intermittent manual hyperinflation or intermittent positive pressure breathing without significantly scaling up the care setting. Furthermore, the detection of suboptimal $P_E$max may allow for the initiation of management strategies to stabilize the patient's condition and prevent future reinstitutions. Compared with inspiratory muscle training [19, 20], few studies have explored expiratory muscle training in terms of prolonged weaning success [21, 22]. As this study is retrospective, we recommend further prospective investigations about the clinical usability of serial and even continuous monitoring of respiratory mechanics during the period before the patients are transferred out of a critical care setting.

There are several limitations to this study. First, there is potential bias due to the single-center retrospective study design, such as the exclusion of patients without weaning parameter data, who may have had different clinical and physiologic features from the analysis cohort. Second, the decision to reconnect MV at the general ward was made mainly by attending physicians or residents rather than intensivists or respiratory therapists. However, as we showed, these reconnected patients had a poor prognosis compared to those who were not reconnected. Therefore, we believe this routine care practice reflected actual patient conditions in that the need for MV reinstitution was made from an intensivist's point of view. Third, we only assessed tracheostomized patients with prolonged MV; therefore, the condition of the patients with reinstitution more than 5 days after they had been extubated could not be analyzed. However, measurements of physiologic parameters in non-tracheostomized patients could be more complicated than in tracheostomized patients once they have been liberated from a ventilator. Fourth, we did not explore the specific clinical conditions of these patients at general wards after they had been transferred out of the weaning unit; therefore, it was difficult to assess the effect of newly developed non-respiratory events on recurrent respiratory failure. Fifth, we did not perform a follow-up study of the patients who were discharged MV free; therefore, the possibility of recurrent MV could not be excluded in the outpatient setting. Sixth, as this was a retrospective study and we retrieved real-world data from the medical records, we were not able to obtain rigorous measurement data within a fixed short interval before the patients proceeded to whole-day unassisted breathing. Although we consider that the weaning parameters over several days might be able to represent the respiratory muscle condition through the course of unassisted breathing trials, prospective studies examining the evolution of weaning parameters before and during the weaning process is needed for the generalization of our findings. Last, the definition of reinstitution might require further study, based on further evidence relating to the relevance of patient status upon transfer to general ward and the event leading to MV reinstitution. The criteria of 60-day reinstitution, despite including most except for three patients, still need further validation.

## Conclusions

In conclusion, there was a high probability of the reinstitution of MV at a general ward despite successful protocoled weaning at the weaning unit in this study, with poor weaning outcomes at a general ward. $P_E$max measured before the unassisted breathing trial was significantly associated with the risk of reinstituting MV at a general ward. Further studies are needed to confirm the relevance of expiratory muscle strength and cough function with regards to patient outcomes after they have been successfully weaned based on the operational criteria.

## Supporting information

**S1 File. Supplementary information.** Additional information regarding the results of the study.
(DOCX)

**S2 File. Patient data.** Data of the cases included in the analysis of this study.
(XLSX)

## Acknowledgments

The authors thank Shu-Hui Yang, Bao-Lin Chang and Li-Min Lin, Department of Nursing, National Taiwan University Hospital, and Jui-Chen Cheng, Department of Integrated Diagnostics & Therapeutics, National Taiwan University Hospital, for their kind assistance in the preparation of clinical data. The authors would like to thank Alfred Hsing-Fen Lin for statistical assistance during the revision phase of this manuscript.

## Author Contributions

**Conceptualization:** Shwu-Jen Lin, Jih-Shuin Jerng, Yao-Wen Kuo, Huey-Dong Wu.

**Data curation:** Shwu-Jen Lin, Chao-Ling Wu.

**Formal analysis:** Shwu-Jen Lin, Jih-Shuin Jerng, Chao-Ling Wu.

**Methodology:** Shwu-Jen Lin, Jih-Shuin Jerng, Yao-Wen Kuo, Shih-Chi Ku, Huey-Dong Wu.

**Project administration:** Jih-Shuin Jerng.

**Resources:** Jih-Shuin Jerng, Chao-Ling Wu, Shih-Chi Ku, Huey-Dong Wu.

**Supervision:** Jih-Shuin Jerng, Shih-Chi Ku, Huey-Dong Wu.

**Validation:** Shwu-Jen Lin, Jih-Shuin Jerng, Yao-Wen Kuo, Chao-Ling Wu.

**Writing – original draft:** Shwu-Jen Lin, Jih-Shuin Jerng.

**Writing – review & editing:** Jih-Shuin Jerng, Yao-Wen Kuo, Shih-Chi Ku, Huey-Dong Wu.

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
