## [Decision Letter · Decision Letter 0]

30 Oct 2019

PONE-D-19-28222

Maximal expiratory pressure is associated with reinstitution of mechanical ventilation after successful unassisted breathing trials in tracheostomized patients with prolonged mechanical ventilation

PLOS ONE

Dear Dr. Jerng,

Thank you for submitting your manuscript to PLOS ONE. After careful consideration, we feel that it has merit but does not fully meet PLOS ONE’s publication criteria as it currently stands. Therefore, we invite you to submit a revised version of the manuscript that addresses the points raised during the review process.

Both reviewers raised several concerns, especially regarding the statistical analysis, data reporting, and interpretations. The authors need to effectively respond to these comments in their revised manuscript.

We would appreciate receiving your revised manuscript by Dec 14 2019 11:59PM. To enhance the reproducibility of your results, we recommend that if applicable you deposit your laboratory protocols in protocols.io, where a protocol can be assigned its own identifier (DOI) such that it can be cited independently in the future. For instructions see: http://journals.plos.org/plosone/s/submission-guidelines#loc-laboratory-protocols

We look forward to receiving your revised manuscript.

Kind regards,

Yu Ru Kou, PhD

Academic Editor

PLOS ONE

Journal Requirements:

Reviewers' comments:

Reviewer's Responses to Questions

**Comments to the Author**

1. Is the manuscript technically sound, and do the data support the conclusions?

Reviewer #1: Partly

Reviewer #2: No

2. Has the statistical analysis been performed appropriately and rigorously? 

Reviewer #1: Yes

Reviewer #2: No

3. Have the authors made all data underlying the findings in their manuscript fully available?

Reviewer #1: Yes

Reviewer #2: Yes

4. Is the manuscript presented in an intelligible fashion and written in standard English?

Reviewer #1: Yes

Reviewer #2: Yes

5. Review Comments to the Author

Reviewer #1: Manuscript: D-19-28222

Title: Maximal expiratory pressure is associated with reinstitution of mechanical ventilation after successful unassisted breathing trials in tracheostomized patients with prolonged mechanical ventilation

Abstract:

The authors retrospectively investigated the occurrence and consequence of MV reinstitution and its associated factors in tracheostomized patients after successful unassisted breathing trials and discharged from RCC. The authors compared the study group (with MV reinstitution) with the control group (without reinstitution before hospital discharge)

Major comments:

1. In Abstract, the statement of “…similar PImax, tidal volume, minute ventilation, respiratory rate, and rapid shallow breathing index before unassisted breathing trial with those without reinstitution.” need to be reedited to “compared with those without reinstitution”.

2. In abstract, the authors reported the results of Cox regression analysis for detecting the factors associated with MV reinstitution within 60 days in hospital. There are two criteria (MV reinstitution before discharge and within 60 days in hospital) to categorize the study patients into the study and control group. Which one is the end-point for Cox regression analysis?

3. In conclusion, the authors declare that “Interventions to enhance expiratory strength in tracheostomized patients are warranted.”. However, in the current study, the study design is retrospective observational and there is no intervention to prove this concept.

4. In Data Collection, the timing for weaning parameters measurement is not clear and consistent during daily protocolized weaning process. Is it measured for (1). initiating unassisted breathing for 12 hours for 2 consecutive days during screening period, (2).direct liberation trial with UBT for 5 days, (3). stepwise liberation, (4). every 14 days during the RCC stay, (5). before patients transferred to ordinary ward. If patients had several measurements of weaning parameter for different purpose, which one is selected to be analyzed?

5. In Materials and Methods, there is no primary outcome measurement mentioned. There is no study design for how to group patients. If the study patient is expired, how to define the group for the case?

6. In Results, the authors reported that the median interval for reinstitution of MV was 8.0days, and 43.2% reinstitutions occurred in <7days after transfer to the general ward. The mean length of stay in RCC was 15.5days, and the authors investigate the reinstitution of MV after RCC discharge. The conditions on RCC discharge have a critical determinant on MV reinstitution in ordinary ward. However, only ICU conditions are showed on Table 1 and analyzed for reinstitution of MV in ordinary ward.

7. In Results, the authors reported Table 1S for 290 cases with weaning parameters measured after successful weaning upon transfer out to a general ward. What is the reason to do it in clinical practice?

8. In Table 3, the method 1 and method 2 are performed for MV reinstitution before discharge and 60 days before discharge. If the study case is connected to MV after RCC discharge, but liberation from MV before hospital discharge. How to define this patient? This classification in Table 3 (within 60 days) for the study patients is different from that in Table 1 (Reinstitution and non-reinstitution before discharge).

9. In table 1, the MV days before RCC transfer is not presented.

10. In Discussion, paragraph 3, the authors mentioned the factors associated with unsuccessful weaning. However, in the current study, the targeted study population is the ones with the reinstitution of mechanical ventilation after successful unassisted breathing trials.

11. In Discussion, paragraph 4, the authors mentioned that “although the reference value for PEmax has not been as frequently discussed in the literature as PImax “. In this study, the values of PEmax in non-reinstituion and reinstitution group are 43.5cmH2O and 37.4 cmH2O (p = 0.001). Is the 6 cmH2O difference of PEmax clinically meaningful?

12. In Discussion, the paragraph 5 and 6 are redundant, Please reedit the content of discussion to focus on the main findings (factors associated with reinstituion of MV) in the current study compared to previous literature.

13. In figure 2, it seems rapid reinstitution of MV after RCC discharge in Pemax <=30cmH2O compared to Pemax >30cmH2O. why is the cut-off value of 30cmH2O is selected and how to explain it?

Reviewer #2: I have read with interest the article entitled “Maximal expiratory pressure is associated with reinstitution of mechanical ventilation after successful unassisted breathing trials in tracheostomized patients with prolonged mechanical ventilation” by Lin et al. This is an interesting topic, which has been poorly explored in the current literature. However, I have comments that need to be addressed before further proceeding.

Major comments.

1) Abstracts. Check the instructions for authors. I don’t think it is relevant to describe the univariate analysis results in this section. Please provide data about outcome, which are missing.

2) Statistical analysis. Cox Model. Can you add on which parameters you adjusted the model, and how these factors were selected? Since there is an important imbalance between groups, this aspect is paramount.

3) Can you provide SAPS II or APACHE at ICU admission? If baseline severity between groups is different, this point could explain in itself the difference in the outcomes between groups.

4) Methodology. Can the authors describe their selection of patients with paired sets of data? I don’t exactly understand what is meant by “paired data sets”? This issue appears only in the results and is not described elsewhere. There is a potential selection bias that needs to be clearly addressed in the methodology: there are less cardiac co-morbidities and neurologic insults in Group B. This issue must be further explained and detailed in the statistical analysis section: how did the authors cope with such issue?

5) Please provide OR, confidence of Interval and p value for multivariable analysis

6) Authors must describe the variable selection process regarding logistic regression and Cox model. I believe that a methodologist should help in the process

7) The MV duration > 30 days as a protective factor of weaning failure is counter-intuitive and not in line with previous data (Beduneau AJRCCM, Funk ERJ) showing that the duration of MV is a risk factor of weaning failure. Can the authors attempt to explain this finding?

8) I am unsure that the logistic regression model obeys the parcimonious rule (ie 1 variable in the model per 5-10 events maximum). The same question arises for the Cox model. Again, a statistical reviewing seems mandatory. Moreover, these models seem redundant (same risk factors)? What is the new information provided by the Cox model? Did the authors study the same outcome?

9) In the end of the results section, the authors provide new data in a 290 patients’ sample with paired data. What is the difference with the previous sample? These data must be clearly expressed in the Statistical analysis section.

10) An effort should be made, in order to reorganize the results section

Minor comments.

1) I believe an English editing, by a native English speaker is mandatory

2) The introduction should be more straightforward. The discussion section must be shortened.

6. PLOS authors have the option to publish the peer review history of their article (what does this mean?). If published, this will include your full peer review and any attached files.

Reviewer #1: Yes: Hsin-Kuo Ko

Reviewer #2: No

---

## [Author Response · Author response to Decision Letter 0]

28 Dec 2019

Responses to Reviewer 1 Comments

Major comments:

1. In Abstract, the statement of “…similar PImax, tidal volume, minute ventilation, respiratory rate, and rapid shallow breathing index before unassisted breathing trial with those without reinstitution.” need to be reedited to “compared with those without reinstitution”.

Response: We have revised this sentence by adding ‘compared’ to it as suggested. Please refer to Abstract of the revised manuscript for the change.

2. In abstract, the authors reported the results of Cox regression analysis for detecting the factors associated with MV reinstitution within 60 days in hospital. There are two criteria (MV reinstitution before discharge and within 60 days in hospital) to categorize the study patients into the study and control group. Which one is the end-point for Cox regression analysis?

Response: To avoid confusion, we have revised the manuscript to focus on “in-hospital MV reinstitution within 60 days” (referred as ‘reinstitution’) as the primary end-point of this study. Therefore, all of the results were rearranged for this end-point. For Cox regression, we only show the result related to this “in-hospital MV reinstitution within 60 days” end-point. Please refer to page 9 of Methods, Table 3 and Results sections the revised manuscript for the changes.

3. In conclusion, the authors declare that “Interventions to enhance expiratory strength in tracheostomized patients are warranted.”. However, in the current study, the study design is retrospective observational and there is no intervention to prove this concept.

Response: We have revised this section and replaced the original sentence with ‘Further studies are needed to confirm the relevance of expiratory muscle strength and cough function with regards to patient outcomes after they have been successfully weaned based on the operational criteria.’ Please refer to Conclusions on Page 21 of the revised manuscript for the change.

4. In Data Collection, the timing for weaning parameters measurement is not clear and consistent during daily protocolized weaning process. Is it measured for (1). initiating unassisted breathing for 12 hours for 2 consecutive days during screening period, (2).direct liberation trial with UBT for 5 days, (3). stepwise liberation, (4). every 14 days during the RCC stay, (5). before patients transferred to ordinary ward. If patients had several measurements of weaning parameter for different purpose, which one is selected to be analyzed?

Response: The timing for weaning parameter measurement was always before the initiation of unassisted breathing trial (UBT). We have added a description to explain that in this study, as the timing of initiating UBT was determined by the clinicians not necessarily linked to the measurement of weaning parameters, we only selected the data of weaning parameters most close to the beginning of UBT. We have provided the information in the description of the Results section that the measurement of parameters was performed on 9.7±7.9 days before the first day of continuous UBT. Please refer to page 8 of Methods section and page 13 of Results section for the change.

5. In Materials and Methods, there is no primary outcome measurement mentioned. There is no study design for how to group patients. If the study patient is expired, how to define the group for the case?

Response: As we have above responded, we have revised the manuscript to focus on “in-hospital MV reinstitution within 60 days” as the primary end-point of this study. A patient was allocated to the “non-reinstitution” group if the patient died in the hospital without reinstitution of MV. Please refer to page 9 of Methods, Table 3 and Results sections the revised manuscript for the changes.

6. In Results, the authors reported that the median interval for reinstitution of MV was 8.0days, and 43.2% reinstitutions occurred in <7days after transfer to the general ward. The mean length of stay in RCC was 15.5days, and the authors investigate the reinstitution of MV after RCC discharge. The conditions on RCC discharge have a critical determinant on MV reinstitution in ordinary ward. However, only ICU conditions are showed on Table 1 and analyzed for reinstitution of MV in ordinary ward.

Response: We have retrieved relevant physiologic data for the calculation of APACHE II scoring for patients on the day they were transferred out of the RCC. The calculated data for APACHE II at RCC discharge as well as vital signs are added to Table 1. With the multivariable Cox regression, we have included the APACHE II scores on ICU admission, on RCC admission and at RCC discharge for the analysis, and the results, as shown in Table 3, did not show a significant association with the risk of MV reinstitution. Please refer to Table 1 and page 15 in Results for the changes.

7. In Results, the authors reported Table 1S for 290 cases with weaning parameters measured after successful weaning upon transfer out to a general ward. What is the reason to do it in clinical practice?

Response: The real-world practice in our institution might include the measurement of weaning parameters for tracheostomized before they are transferred out the RCC after days of un-interrupted UBT, in case there is a concern for reconnection to MV at the general ward; however, the decision for measuring was based on the judgement of clinicians and respiratory therapist, who might provide expert opinions about further care at the general wards. We have added this description to the Methods section. There was no significant evolutional change in the data for each parameter of these 290 patients (Table 2S); therefore, we decided to include only weaning parameters before the unassisted breathing trial for consideration of multivariable Cox regression analysis. Please refer page 9 in the Methods section for the change.

8. In Table 3, the method 1 and method 2 are performed for MV reinstitution before discharge and 60 days before discharge. If the study case is connected to MV after RCC discharge, but liberation from MV before hospital discharge. How to define this patient? This classification in Table 3 (within 60 days) for the study patients is different from that in Table 1 (Reinstitution and non-reinstitution before discharge).

Response: The outcomes regarding MV reinstitution at the general ward were described in the Results on pages 10-11. For better display of the information, we have rearranged the Results and Tables. We have revised Tables 1 and 2 to show the comparison of patients with and without in-hospital MV reinstitution within 60 days, whereas the patients who had reinstitution of MV were allocated to the ‘reinstitution’ group. In Table 3, we now only show the results for Cox regression using the time-dependent outcome of “in-hospital reinstitution within 60 days” to avoid confusion to the readers. Please refer to pages 10-11, Tables 1, 2 and 3 for the changes.

9. In table 1, the MV days before RCC transfer is not presented.

Response: To Table 1 we have added the data relevant to MV use, including total days of MV use, ICU MV days, and RCC MV days. These data were similar between the reinstitution and non-reinstitution groups of analyses. Please refer to Table 1 in the Results section for the change.

10. In Discussion, paragraph 3, the authors mentioned the factors associated with unsuccessful weaning. However, in the current study, the targeted study population is the ones with the reinstitution of mechanical ventilation after successful unassisted breathing trials.

Response: As there was limit reports in the literature regarding the factors associated with MV reinstitution in tracheostomized patients with prolonged MV, we have shortened this paragraph to avoid confusion to the readers. Please refer to the Discussion section of the revised manuscript for the change.

11. In Discussion, paragraph 4, the authors mentioned that “although the reference value for PEmax has not been as frequently discussed in the literature as PImax “. In this study, the values of PEmax in non-reinstitution and reinstitution group are 43.5cmH2O and 37.4 cmH2O (p = 0.001). Is the 6 cmH2O difference of PEmax clinically meaningful?

Response: As this study was conducted retrospectively, we recognized that the patient population in this study was heterogenous that multiple factors affected the outcomes. The data in Results and Table 3 support that factors other than PEmax affected the outcome independently, such as co-morbidities, complications at the ICU, severity of organ dysfunction at the ICU, and MV duration. Therefore, the univariate comparison of PEmax between these two groups (reinstitution vs non-reinstitution) might not infer the independent significance of association, or preclude this factor for multivariable analysis.

12. In Discussion, the paragraph 5 and 6 are redundant, Please reedit the content of discussion to focus on the main findings (factors associated with reinstitution of MV) in the current study compared to previous literature.

Response: We have reduced the content of these paragraphs to make the Discussion more concise. Please refer to Discussion of the revised manuscript for the changes.

13. In figure 2, it seems rapid reinstitution of MV after RCC discharge in Pemax <=30cmH2O compared to Pemax >30cmH2O. why is the cut-off value of 30cmH2O is selected and how to explain it?

Response: There were limited reports in the literature regarding the reference PEmax value for predicting outcomes for tracheostomized patients. For clinical practice, we also chose to test the discriminating power by the nearest ten value; therefore, we have also performed Cox regression analysis to use PEmax > 20 cmH2O to replace PEmax > 30 cmH2O as the cut-off value, and found that the protective significance for MV reinstitution was lost; therefore, we chose PEmax > 30 cmH2O as the cut-off value. We also addressed this point in the ‘Statistical analysis’ subsection of the ‘Methods’ section.

 

Responses to Reviewer 2 Comments

Major comments.

1) Abstracts. Check the instructions for authors. I don’t think it is relevant to describe the univariate analysis results in this section. Please provide data about outcome, which are missing.

Response: We have revised the Abstract section to remove the univariate analysis and add outcome descriptions. Please refer to the revised manuscript for the change.

2) Statistical analysis. Cox Model. Can you add on which parameters you adjusted the model, and how these factors were selected? Since there is an important imbalance between groups, this aspect is paramount.

Response: As we had found that the patient population was very heterogeneous, therefore we selected the variables with a p-value of less than 0.1 into the multivariable Cox regression. As we have re-performed the Cox regression analysis, the final model now contains 13 variables, with basically PEmax > 30 cmH2O remaining as significant protecting factor for MV reinstitution. Please refer to the revised manuscript at Table 3. We also addressed how to select these factors in the ‘Statistical analysis’ subsection of the ‘Methods’ section.

3) Can you provide SAPS II or APACHE at ICU admission? If baseline severity between groups is different, this point could explain in itself the difference in the outcomes between groups.

Response: We have added the results of APACHE II to Table 1, which now contains APACHE II scores on ICU admission, on admission to RCC and at RCC discharge, and the comparisons between the reinstitution and non-reinstitution groups. Please refer to the Results and Table 1 in the revised manuscript. Although the reinstitution group had higher APACHE II scores on ICU admission, on admission to RCC and at RCC discharge scores in univariable analysis, the multivariable Cox analysis showed that none of them were independent factors of risk of reinstitution of MV.

4) Methodology. Can the authors describe their selection of patients with paired sets of data? I don’t exactly understand what is meant by “paired data sets”? This issue appears only in the results and is not described elsewhere. There is a potential selection bias that needs to be clearly addressed in the methodology: there are less cardiac co-morbidities and neurologic insults in Group B. This issue must be further explained and detailed in the statistical analysis section: how did the authors cope with such issue?

Response: The real-world practice in our institution might include the measurement of weaning parameters for tracheostomized before they are transferred out the RCC after days of un-interrupted UBT, in case there is a concern for reconnection to MV at the general ward; however, the decision for measuring was based on the judgement of clinicians and respiratory therapist, who might provide expert opinions about further care at the general wards. We have added this description to the Methods section. As further statistical analysis did not show new findings, we decided to remove the statement of statistical analysis for the weaning parameter measurement at RCC discharge. Please refer page 9 in the Methods section for the change.

5) Please provide OR, confidence of Interval and p value for multivariable analysis

Response: As in Table 3 the 95% CIs and p values have been provided, we have added hazard ratio (HR), 95%CI and p values to the Abstract for Cox regression. Please refer to the Abstract of revised manuscript for the change.

6) Authors must describe the variable selection process regarding logistic regression and Cox model. I believe that a methodologist should help in the process

Response: As has been provided in the response to Comment #2, we have provided the description of variable selection based on our initial findings. Please refer to our response to Comment #2 and the ‘Statistical analysis’ subsection of the ‘Methods’ section. To avoid confuse the readers, we decided to obtain the result of Cox model and remove the logistic regression version. We did actually consult a senior statistician and we have also acknowledged for his help in the ‘Acknowledgments’ section of the revised manuscript

7) The MV duration > 30 days as a protective factor of weaning failure is counter-intuitive and not in line with previous data (Beduneau AJRCCM, Funk ERJ) showing that the duration of MV is a risk factor of weaning failure. Can the authors attempt to explain this finding?

Response: We agree that previous reports, mainly focusing on overall weaning prognosis assessment, concluded that prolonged MV > 30 days as a negative predictor for successful weaning. However, as the revised Cox regression analysis did not show MV > 30 days as a significant factor, we decide not to include this factor in the discussion. Please refer to Table 3 of the revised manuscript.

8) I am unsure that the logistic regression model obeys the parsimonious rule (ie 1 variable in the model per 5-10 events maximum). The same question arises for the Cox model. Again, a statistical reviewing seems mandatory. Moreover, these models seem redundant (same risk factors)? What is the new information provided by the Cox model? Did the authors study the same outcome?

Response: As we have rearranged the results to focus on 60-day in-hospital MV reinstitution, we have reduced the descriptions of multivariable analysis to only Cox regression analysis. The process of choosing variables is provided in the ‘Statistical analysis’ subsection of the ‘Methods’ section.

A total of 13 candidate predictors were introduced into the multivariable Cox regression and we had 116 events (reinstitution of MV within 60 days), therefore the final presentation in Table 3 would conform to the parsimonious rule. Please refer to the Methods section and Table 3 of the revised manuscript for the changes. In addition, as mentioned at comment#7, we did actually consult a senior statistician and we have also acknowledged for his help in the ‘Acknowledgments’ section of the revised manuscript

9) In the end of the results section, the authors provide new data in a 290 patients’ sample with paired data. What is the difference with the previous sample? These data must be clearly expressed in the Statistical analysis section.

Response: We have added descriptions regarding the methods for assessing the paired data mentioned in the Supplement materials. There was no significant evolutional change in the data for each parameter of these 290 patients (Table 2S); therefore, we decided to include only weaning parameters before the unassisted breathing trial for consideration of multivariable regression analysis. Please refer to the Methods sections of the revised manuscript for the changes.

10) An effort should be made, in order to reorganize the results section

Response: We have reorganized the Results section for better readability. Please refer to the Results sections of the revised manuscript for the changes.

Minor comments.

1) I believe an English editing, by a native English speaker is mandatory

Response: The manuscript had actually been edited by a native English-speaker. We have repeated this English editing process for the revised manuscript. 

2) The introduction should be more straightforward. The discussion section must be shortened.

Response: We have thereby revised these sections to provide more readability. Please refer to the Introduction and Discussion sections of the revised manuscript for the changes.

---

## [Decision Letter · Decision Letter 1]

14 Jan 2020

PONE-D-19-28222R1

Maximal expiratory pressure is associated with reinstitution of mechanical ventilation after successful unassisted breathing trials in tracheostomized patients with prolonged mechanical ventilation

PLOS ONE

Dear Dr. Jerng,

Thank you for submitting your manuscript to PLOS ONE. After careful consideration, we feel that it has merit but does not fully meet PLOS ONE’s publication criteria as it currently stands. Therefore, we invite you to submit a revised version of the manuscript that addresses the points raised during the review process.

Both reviewers continued to raise some concerns. The reviewer 1 is very critical and the authors must effectively respond to his/her comments. Please note that the final acceptance of the submission needs to get the approval from both reviewers.

We would appreciate receiving your revised manuscript by Feb 28 2020 11:59PM. To enhance the reproducibility of your results, we recommend that if applicable you deposit your laboratory protocols in protocols.io, where a protocol can be assigned its own identifier (DOI) such that it can be cited independently in the future. For instructions see: http://journals.plos.org/plosone/s/submission-guidelines#loc-laboratory-protocols

We look forward to receiving your revised manuscript.

Kind regards,

Yu Ru Kou, PhD

Academic Editor

PLOS ONE

Reviewers' comments:

Reviewer's Responses to Questions

**Comments to the Author**

1. If the authors have adequately addressed your comments raised in a previous round of review and you feel that this manuscript is now acceptable for publication, you may indicate that here to bypass the “Comments to the Author” section, enter your conflict of interest statement in the “Confidential to Editor” section, and submit your "Accept" recommendation.

Reviewer #1: All comments have been addressed

Reviewer #2: All comments have been addressed

2. Is the manuscript technically sound, and do the data support the conclusions?

Reviewer #1: Partly

Reviewer #2: Yes

3. Has the statistical analysis been performed appropriately and rigorously? 

Reviewer #1: Yes

Reviewer #2: Yes

4. Have the authors made all data underlying the findings in their manuscript fully available?

Reviewer #1: Yes

Reviewer #2: Yes

5. Is the manuscript presented in an intelligible fashion and written in standard English?

Reviewer #1: Yes

Reviewer #2: Yes

6. Review Comments to the Author

Reviewer #1: 1. The authors revised the in-hospital MV reinstitution within 60 days” (referred as ‘reinstitution’) as the primary end-point of this study. Why is the 60 days determined to evaluate the ventilator outcome? The PEmax before unassisted breathing trial predicts the 60-day outcome of hospitalization. Is it reasonable?

2. About the issue of weaning parameters measurement, the authors replied that “We have added a description to explain that in this study, as the timing of initiating UBT was determined by the clinicians not necessarily linked to the measurement of weaning parameters, we only selected the data of weaning parameters most close to the beginning of UBT.”. The timing of UBT is determined by physicians. The parameters measurement and UBT are two separately events during ventilator care, and a variable and prolonged period was noted (9.7±7.9 days). It is quite difficult for physicians to apply the results of the current study on patient care. The main problem is which one of measured PEmax close to UBT is unable to be determined prospectively. Before the initiation of UBT determined by physicians, whether the PEmax measured by RT today would be the one to predict the future 60-day ventilator outcome remains unknown.

Reviewer #2: I believe my comments have been adequately addressed. I have minor 3 comments:

-There are numerous abbreviations (PMV, UBT etc...). This is a bit misleading for the reader and I don't thinnk it is mandatoy ot have so many. Please check this aspect in order to improve the lisibility.

-Please chekc the instructions for authors. In the Tables, the p-values have many numbers after the coma. I believe that false accuracy should be avoided.

-I believe the Discussion is still too long and should be shortened.

7. PLOS authors have the option to publish the peer review history of their article (what does this mean?). If published, this will include your full peer review and any attached files.

Reviewer #1: No

Reviewer #2: No

---

## [Author Response · Author response to Decision Letter 1]

8 Feb 2020

Responses to Reviewer 1 Comments

1. The authors revised the in-hospital MV reinstitution within 60 days” (referred as ‘reinstitution’) as the primary end-point of this study. Why is the 60 days determined to evaluate the ventilator outcome? The PEmax before unassisted breathing trial predicts the 60-day outcome of hospitalization. Is it reasonable?

Response: Thank you for the comments. Our rationale for the primary outcome was that if a patient was reconnected to the ventilator before discharge, the patient would then need a care setting especially for the use of mechanical ventilation, and this might have a significant influence on the burden of care and patient prognosis. In addition, there was a gradual reduction in the chance to reinstitute MV (page 10 of the Results section), but it was difficult to find a time earlier than 60 days to determine the non-institution group, as this could have resulted in bias due to dichotomized grouping. Therefore, we originally preferred to include all cases of in-hospital reinstitution of MV as the primary outcome. However, as some of the patients in this study cohort actually stayed in the hospital for a period beyond 60 days, in whom the influence of physiologic status upon RCC discharge may have been uncertain even if they were later reinstituted, we chose 60 days as a censor time point. In fact, only three patients reinstituted MV beyond 60 days as they remained hospitalized (who were allocated as the non-reinstitution group in this study, see Page 11), and we considered that this may be reasonable given that the 60-day censor time included most of the patients based on our rationale, and that this cut-off time may have resulted in the least influence on grouping bias of reinstitution. Please refer to the Discussion section of the revised manuscript for this limitation.

2. About the issue of weaning parameters measurement, the authors replied that “We have added a description to explain that in this study, as the timing of initiating UBT was determined by the clinicians not necessarily linked to the measurement of weaning parameters, we only selected the data of weaning parameters most close to the beginning of UBT.”. The timing of UBT is determined by physicians. The parameters measurement and UBT are two separately events during ventilator care, and a variable and prolonged period was noted (9.7±7.9 days). It is quite difficult for physicians to apply the results of the current study on patient care. The main problem is which one of measured PEmax close to UBT is unable to be determined prospectively. Before the initiation of UBT determined by physicians, whether the PEmax measured by RT today would be the one to predict the future 60-day ventilator outcome remains unknown.

Response: It is our understanding that weaning parameters in tracheostomized patients with prolonged MV can be used for multiple purposes, such as the decision on whether to initiate active lowering of MV settings, and the speed and tempo for protocoled weaning involving unassisted breathing trials interspersed with MV support. Therefore, the measurement of weaning parameters usually took place several days before the patient actually started the whole-day unassisted breathing trials for 5 successive days. As this was a retrospective study and we retrieved real-world data from the medical records, we were not able to obtain rigorous measurement data within a fixed short interval before the patients proceeded to whole-day unassisted breathing. In addition, we compared the weaning parameters before and after whole-day unassisted breathing trials in 290 patients (64% of the study cohort) with valid paired data, as describe on Page 14, and found that the available paired weaning parameters did not show significant changes even after the patients were deemed to have been successfully liberated from the ventilator. Although we consider that the weaning parameters over several days might be able to represent the respiratory muscle condition through the course of unassisted breathing trials, we agree that a prospective study examining the evolution of weaning parameters before and during the weaning process is needed for the generalization of our findings. Please refer to the Discussion section of the revised manuscript for the limitations of this study.

Thank you so much for your very constructive opinions and comments, these points have greatly enhanced the quality of our manuscript. 

 

Responses to Reviewer 2 Comments

1. There are numerous abbreviations (PMV, UBT etc...). This is a bit misleading for the reader and I don't think it is mandatory to have so many. Please check this aspect in order to improve the lisibility.

Response: Thank you for your comments. We have revised the manuscript to reduce the use of abbreviations. Please refer to the revised manuscript for the changes.

2. Please check the instructions for authors. In the Tables, the p-values have many numbers after the coma. I believe that false accuracy should be avoided.

Response: We have revised the Table and Results section to reduce the p-values from 3 digits to 2 digits after the comma. Please refer to the revised manuscript for the changes.

3. I believe the Discussion is still too long and should be shortened.

Response: We have further shortened the Discussion section (from 1,305 to 1,101 words) to improve the readability and clarity. Please refer to the Discussion section of the revised manuscript for the changes.

Thank you for your constructive suggestions. We think that they have greatly enhanced the quality of our manuscript.

---

## [Decision Letter · Decision Letter 2]

19 Feb 2020

Maximal expiratory pressure is associated with reinstitution of mechanical ventilation after successful unassisted breathing trials in tracheostomized patients with prolonged mechanical ventilation

PONE-D-19-28222R2

Dear Dr. Jerng,

We are pleased to inform you that your manuscript has been judged scientifically suitable for publication and will be formally accepted for publication once it complies with all outstanding technical requirements.

With kind regards,

Yu Ru Kou, PhD

Academic Editor

PLOS ONE

Additional Editor Comments (optional):

Reviewers' comments:

Reviewer's Responses to Questions

**Comments to the Author**

1. If the authors have adequately addressed your comments raised in a previous round of review and you feel that this manuscript is now acceptable for publication, you may indicate that here to bypass the “Comments to the Author” section, enter your conflict of interest statement in the “Confidential to Editor” section, and submit your "Accept" recommendation.

Reviewer #1: All comments have been addressed

Reviewer #2: All comments have been addressed

2. Is the manuscript technically sound, and do the data support the conclusions?

Reviewer #1: Yes

Reviewer #2: Yes

3. Has the statistical analysis been performed appropriately and rigorously? 

Reviewer #1: Yes

Reviewer #2: Yes

4. Have the authors made all data underlying the findings in their manuscript fully available?

Reviewer #1: Yes

Reviewer #2: Yes

5. Is the manuscript presented in an intelligible fashion and written in standard English?

Reviewer #1: Yes

Reviewer #2: Yes

6. Review Comments to the Author

Reviewer #1: The authors have completely responded to my comments. No further revision is needed to improve the quality for publication.

Reviewer #2: Dear authors,

I have no further comments about this article. I believe all issues have been addressed.

Best regards

7. PLOS authors have the option to publish the peer review history of their article (what does this mean?). If published, this will include your full peer review and any attached files.

Reviewer #1: No

Reviewer #2: No

---

## [Editor Report · Acceptance letter]

21 Feb 2020

PONE-D-19-28222R2 

Maximal expiratory pressure is associated with reinstitution of mechanical ventilation after successful unassisted breathing trials in tracheostomized patients with prolonged mechanical ventilation 

Dear Dr. Jerng:

I am pleased to inform you that your manuscript has been deemed suitable for publication in PLOS ONE. Congratulations! Your manuscript is now with our production department. 

With kind regards,

on behalf of

Dr. Yu Ru Kou 

Academic Editor

PLOS ONE